



# Scattered coal is the largest source of ambient volatile organic compounds during the heating season in Beijing

Yuqi Shi[1], Ziyan Xi[1], Maimaiti Simayi[1], Jing Li[1,2], Shaodong Xie[1]

[1]College of Environmental Sciences and Engineering, State Key Joint Laboratory of Environmental Simulation and Pollution Control, Peking University, Beijing, 100871, PR China

[2]Department of Environmental Health, Harvard T.H. Chan School of Public Health, Boston, 02215, USA

*Correspondence to*: Shaodong Xie (sdxie@pku.edu.cn)

**Abstract.** We identified scattered coal burning as the largest contributor to ambient volatile organic compounds (VOCs), exceeding traffic-related emissions, during the heating season in Beijing prior to the rigorous emission limitations enacted in 2017. However, scattered coal is underestimated in emission inventories generally, because the activity data are incompletely recorded in official energy statistics. Results of positive matrix factorization (PMF) models confirmed that coal burning was the largest contributor to VOC concentrations prior to the emission limitations of 2017, and a reduction in scattered coal combustion, especially in rural residential sector, was the primary factor in the observed decrease in ambient VOCs and secondary organic aerosol (SOA) formation potential in urban Beijing after 2017. Scattered coal burning was included in a corrected emission inventory and we obtained comparable results between this corrected inventory and PMF analyses, particularly for the non-control period. However, a refined source sub-classification showed that passenger car exhaust, petrochemical manufacturing, gas stations, traffic evaporation, traffic equipment manufacturing, painting, and electronics manufacturing are also contributors to ambient VOCs. These sources should focus on future emission reduction strategies and targets in Beijing. Moreover, in other region with scattered coal-based heating, scattered coal burning is still the key factor to improve the air quality in winter.

## 1 Introduction

In Chinese cities, a severe deterioration in air quality has threatened human health (Han et al., 2018). Extremely poor air quality in cities such as Beijing, one of the world's largest and China's capital, is a result of growth in fossil-fuel economies, the expansion of industrial manufacturing, heavy traffic, and large-scale urban construction (Ru et al., 2015;Zeng et al., 2005;Liu

and Wu, 2013;Tang, 2007). Within China, organizations such as the Joint Prevention and Control of Atmospheric Pollution (JPCAP) and Regional Atmospheric Pollution Control (RAPC) have identified volatile organic compounds (VOCs) as key air pollutants (Zhou and Elder, 2013). VOCs are precursors of secondary organic aerosols (SOAs), which are in turn related to particulate matter < 2.5 μg (PM$_{2.5}$), and photochemically produced ozone (O$_3$) (Sillman, 1999;Pandis, 1997). The negative effects of VOCs on human health have received increased public attention (Logue et al., 2010;Zhang et al., 2012). Globally,

natural sources are significant emitters of VOCs, however, anthropogenic sources are far more prevalent in urban areas (Guenther et al., 2006;Janssens-Maenhout et al., 2015), particularly in winter at northern latitudes when biological emissions are low (Li and Xie, 2014). Therefore, it is important to incorporate seasonality into understanding major anthropogenic sources of VOCs and in developing effective controls and mitigation measures for air pollution.

Ambient VOC concentrations in Beijing show significant seasonal variation and are highest during the heating season

(November to March, when centralized district heating is turned on) (Liu et al., 2005;Wang et al., 2014;Wei et al., 2018). Satellite-derived emission inventories have suggested monthly variation in total VOC emissions, with distinct highs during the heating season (Li et al., 2019b). Results of source apportionment analyses have indicated differences in source contributions of VOCs seasonally, with high proportions coming from coal combustion during the heating season and traffic exhaust in the remaining months (Wang et al., 2014;Wang et al., 2013a). However, most emission factor (EF)-based inventories do not

include seasonal variation.

Scattered coal (SC) is defined as those coal of poor quality which is dispersedly used in civilian (households cooking and heating, commercial and public, rural production, etc.), and industry (small-scale industrial boilers and furnaces). The coal of poor quality, with high content of ash, sulfur and volatile matters, is widely used in rural residential sector due to the low price,

besides, it is different to cover those decentralize-used coal overall in official energy statistics (Cheng et al., 2017;Peng et al.,

2019;Huo et al., 2017). Different from efficient centralized coal combustion, such as power generation, heat supply and large-scale industrial boilers, SC burning is a near-ground and non-point source, its low combustion efficiency and air pollution control deficiency results in higher air pollutant emission intensity, which brings negative influence on ambient air and has a more direct adverse impact on the human health (Finkelman et al., 1999).

Especially, as a major ambient pollution source, household energy consumption has attributed a large number of deaths

especially in rural areas (Zhao et al., 2019), and attracts more and more public attention (Liu et al., 2016). It's undeniable that a reduction in conventional household solid fuel consumption has improved air quality and human health in northern China over recent decades (Zhao et al., 2018). Nevertheless, solid fuel combustion remains an important emission source. Moreover, rural residential coal combustion affects not only rural but urban air quality, and has higher contributions in winter especially in northern China for a long time (Shen et al., 2019). However, a large proportion of residential coal use is overlooked in China

Energy Statistical Yearbook (CESY), which means that most EF-based inventories may poorly estimate coal consumption. Using field sampling and remote sensing, many studies have showed that the actual amount of rural and urban coal consumption is much higher than the statistical data in CESY. Peng et al. (2019) conducted a field survey in 2010 to obtain data for solid fuel consumption and use patterns in Chinese counties, and accordingly estimated 62% higher than the coal consumption reported in CESY for the rural residential sector in China. Cheng et al. (2017) summarized the investigated coal

consumption of several studies in rural area, and estimated the residential coal consumption from 1996 to 2014 in the BTH region, which triples coal consumption reported in CESY. Cheng et al. (2016) also estimated Beijing's residential coal combustion at $400 \times 10^4$ t in 2015. Previous research, combined with the prevalence of coal-based heating in northern China (Wang et al., 2013a), suggests that coal combustion is an important contributor to ambient VOCs over winter, but the magnitude of this contribution is not yet understood. Li et al. (2019b) showed similar PMF results in an evaluation of emissions

in the winter of 2015 in Beijing, where fuel combustion contributed >50% of ambient VOCs. They also proposed that the

essential parts of fuel combustion might be the undocumented consumption of coal briquettes and chunks, but did not give any further explicit evidence.

We focused on confirming SC burning as a critical anthropogenic VOC emission source contributing to high VOC concentrations during the heating season in Beijing. We discuss the efficacy of air pollution control periods in light of observed 70 variation in emission intensities. We used the positive matrix factorization (PMF) model to quantify the contributions of different emission sources to observed VOC concentration data. The contribution of coal burning was confirmed by examining variation among emission sources, control measures placed on sources, and emission intensities. We estimated a monthly corrected EF-based inventory for the heating season, and compared these estimates to the PMF results. We calculated secondary organic aerosols potential (SOAP) values for different emission sources based on the results of the PMF and 75 emission inventory to determine the largest contributor to total SOAP reduction. We further evaluate and discuss the control policies of 2017.

## 2 Methods

### 2.1 Site description

Air quality measurements were collected during two consecutive heating seasons, the first from December 2016 to January 80 2017, when air quality control measures were not heavily enforced (non-control period), and the second from December 2017 to January 2018, when air quality control measures were rigorously enforced (control period). Measurements were taken at the fifth story of a building on the Peking University (PKU) campus in northwestern Beijing (39.99°N, 116.33°E), at a height of approximately 12 m. The building was surrounded by several five- or six-story buildings and one side road to the east. This site is located approximately 700 m north of 4th ring road (a major city traffic line) and 10 km from the centre of Beijing (Fig. 85 S1). The surrounding area is primarily commercial and residential, and the major nearby emission source is road vehicles. This site is considered to be representative of a typical urban environment in Beijing.

## 2.2 Sampling and analyses

We used a continuous sampling and analyses method for ambient VOCs, which has been described in detail in previous studies (Wu et al., 2016b;Li et al., 2015a). Automated, hourly sample collection was achieved using a custom-built online GC–MS/FID

system (TH-PKU 300B, Wuhan Tianhong Instrument Co. Ltd., China; GCMS-QP2010SE, Shimadzu, Japan). We applied rigorous quality-assurance (QA) and quality-control (QC) procedures. Daily calibrations were performed every 5 days, and the calibration curve results of each target species with <10% variation were considered acceptable relative to the actual concentrations. The method detection limit (MDL) for each species quantified using this system ranged from 0.01 to 0.10 ppbv. Detailed information is shown in Text S1.

## 2.3 Source apportionment

The USEPA PMF model (version 5.0) has been applied to a wide range of data, including 24 h speciated $PM_{2.5}$, size-resolved aerosols, deposition, air toxins, high-time-resolution measurements such as those from aerosol mass spectrometers (AMSs), and VOC concentrations. As a receptor model, PMF is a mathematical approach. Composition or speciation is determined using analytical methods appropriate for the media. We applied this PMF model to determine source apportionment for

measured ambient VOCs. A PMF requires two input files: a concentration dataset comprising a suite of parameters measured across multiple samples, and an uncertainty dataset comprising uncertainty values for each species and sample. VOC species that were below the MDL in >50% of samples or that showed a very small signal-to-noise ratio (S:N) were omitted from analyses. S:N was calculated for each species using PMF, where the signal represents the difference between concentration and uncertainty.

We categorized VOC species as "strong," "weak," and "bad." Strong was the default value for all species, weak indicates those with tripled uncertainty, and VOCs categorized as bad were removed from further analyses. The final dataset comprised 1,918 samples of 53 compounds (42 strong and 11 weak), which accounted for 90% of the total mixing ratios. Modeling was performed using 4 – 11 factors and the 8-factor solution was deemed to be the most representative. In profile analysis technology, ambient VOC concentration can be considered the linear addition of VOC compositions derived from various

sources. Characteristics of each pollution source can be used, to some extent, to resolve issues with collinearity in the source

component spectrum.

**2.4 VOC emission inventory**

Through a systematic literature review we found that while anthropogenic VOC emission inventory methodologies are similar,

source classification and EFs differ between studies. To promote the comparability of PMF and emission inventory results we

proposed and utilized a modified source classification system based on the existing four-level categorization (Wu et al., 2016a).

Level 1 contains seven sublevels: coal burning, fuel oil and gas, traffic exhaust, petroleum-related evaporation, VOC-related

industry, VOC-product utilization, and biomass-burning. Level 2 represents further divisions of these sublevels. For example,

coal burning was further divided into burning of SC and centralized coal. Level 3 again divided these categories, where

centralized coal burning was divided based on the consumption terminus, such as manufacturing, power generation, or heating.

Level 3 divisions were further split in Level 4 categories based on highly detailed information. The detailed classification

method is provided in Table S1.

VOC emission calculations were EF-based. EF and source profiles of on-road vehicles were calculated using the Computer

Programme to Calculate Emissions from Road Transport version 5 (COPERT 5; https://www.emisia.com/utilities/copert), the

methods for which have been explained in detail in previous studies (Cai and Xie, 2013). Input parameters included vehicle

type, number of vehicles, the average speed and annual mileage of different vehicle types, and monthly ambient temperature,

and mileage degradation of vehicles was considered (Cai and Xie, 2009). Vehicle emissions included tailpipe exhaust and

evaporation, which can be estimated separately by COPERT 5. For other emission sources, we relied on EFs and source

profiles provided in the pre-established emission inventory for the Beijing–Tianjin–Hebei (BTH) region (Bo et al., 2008).

Many studies have estimated the SC consumption recent years, but few of that have estimated SC reductions in 2017 compared

to 2016. In an effort to control coal consumption and promote clean energy sources, the Natural Resources Defense Council

(NRDC), in collaboration with government and other relevant organizations, launched the China Coal Consumption Cap Plan

and Policy Research Project (COALCAP) in October, 2013 (NRDC, 2013). Reports produced by this project provide estimated

reductions of SC consumption in 2017, and reduction proportions among different terminal sectors. Therefore, a synthesis of previous studies (Peng et al., 2019;Cheng et al., 2017;Cheng et al., 2016;Huo et al., 2017) and COALCAP reports provides

the data required to estimate the VOC emissions from SC (both civil and industrial sectors). For monthly profiles, it was assumed that residential, commercial and public SC consumption only occur during the heating season, and consumption was averaged across the season. SC consumption for rural production sector and industrial sector was averaged across the entire year. Monthly activity data and profiles for other sources were obtained as described below.

Except SC, monthly data of other residential energy consumption were estimated based on household survey results (Wu and

Xie, 2018). Most of monthly data for industrial sector emissions were developed based on outputs of industrial products (NBS/BBS). Power plant data were derived from power-generation statistics (NBS). Heat supply data were averaged across the heating season. Monthly distribution of road vehicle emissions was derived from Li et al. (2017). Agricultural burning of crop residue was estimated based on Moderate Resolution Imaging Spectroradiometer (MODIS) fire counts in croplands (Li et al., 2016b). We assumed that emissions from other sources did not vary across months (Wu and Xie, 2018). Corresponding

to the period that ambient concentration data were collected at PKU, we established a monthly emission inventory. The detailed monthly data are provided in Table S5.

**2.5 SOAP contributions of each VOC source**

SOAP method has been widely used for the estimation of SOA formation potential based on emission inventories and observation data (Barthelmie and Pryor, 1997;Wu et al., 2017;Wu and Xie, 2018). Explicit chemical models and SOA yield

models are two accepted methods used to calculate SOAP (Wu et al., 2017). The process of SOA formation from VOCs has been explored and summarized extensively, and is affected by atmospheric or experimental conditions, such as water vapor, temperature, light, organic aerosol concentration, oxidant type, and the concentration of nitrogen oxides (NOx) (Hallquist et al., 2009;Warren, 2008). Complexities and uncertainty in the SOA reaction mechanism creates difficulties in accurately modeling SOA formation in the atmosphere. Therefore, using parameters acquired under similar conditions is advantageous

for regional estimates of SOAP. Here, SOAP-weighted mass contributions, as defined by Derwent et al. (2010) and cited by



many researches (Gilman et al., 2015;Redington and Derwent, 2013;Li et al., 2015a), were used to evaluate precursor source

contributions and variation within different control periods on SOA formation. The definition of this SOAP method describes

the mass of aerosol produced per mass of VOC reacted and expressed relative to toluene, which is different from the absolute

SOA formation potential value (the mass of aerosol formed per mass of VOC reacted).

SOA potentials of this method were simulated under test conditions of high anthropogenic emissions of VOCs and NOx

(Derwent et al., 1998). Due to the low contribution from natural emissions, anthropogenic SOAs predominant in this scenario.

Toluene was chosen as the basic compound for SOAP estimation because of its well-characterized man-made emission status

and importance as an SOA precursor (Ng et al., 2007). The amount of SOAs formed is described using a toluene-equivalent,

and SOAPs of each compound are expressed as an index relative to toluene. The SOAP represents the propensity for an organic

compound to form SOA when an additional mass emission of that compound is added to the ambient atmosphere expressed

relative to that SOA formed when the same mass of toluene is added (Derwent et al., 2010). We hypothesized that all VOC

species would have an effect on SOA formation. SOAP-weighted mass contributions were calculated based on PMF results

(where VOC units were converted from ppbv to $\mu g\ m^{-3}$) and the corrected emission inventory (Gg), respectively. The SOAP-

weighted mass contribution of each VOC source can be calculated using Eq. (1):

$$\text{SOAP}_{weighted\ mass\ contribution} = \sum VOC_i \times SOAP_i \qquad\qquad (1)$$

where $VOC_i$ is the mass contribution of a VOC source to species $i$ ($\mu g\ m^{-3}$ / Gg); $SOAP_i$ is the SOA formation potential for

species $i$ (unitless). Table S2 shows a listing of the propensities for secondary organic aerosol formation expressed on a mass

emitted basis as SOAPs relative to toluene=100 for 113 organic compounds.

This SOAP method removes issues associated with uncertainty in absolute SOA concentrations (Li et al., 2015a). Besides, this

SOAP method is appropriate for conditions of high anthropogenic emissions of VOCs and NOx (Derwent et al., 1998).

Although highly idealized, these conditions are comparable to those in urban Beijing during control and non-control periods.



## 3 Results and Discussion

### 3.1 Unprecedented air pollution control measures in China

In 2013, The People's Republic of China State Council, in determining that improving air quality was not only a human health
issue but was also an important focus of economic growth and security, deployed the Action Plan of Air Pollution Prevention
and Control (the Action Plan) (http://www.gov.cn). Emission control measures implemented in the Beijing Action Plan (2013
– 2017) were summarized by Cheng et al. (2019). The Action Plan mandated that the average annual concentration of $PM_{2.5}$
had to be limited to 60 μg m$^{-3}$ in Beijing, and reduced by over 25%, relative to a 2012 baseline, in BTH by 2017. Since 2013,
further plans and laws, namely the Air Pollution Prevention Law of 2016, were released to curb emissions and meet air-quality
targets. After 3 years of these efforts (2013 – 2016), air-quality improvement was less than satisfactory. Hence, in early 2017,
the Chinese government released Ten Heavier Measures to Prevent and Control Air Pollution (the Ten Measures) in Beijing.
A detailed description of these enhanced control measures is shown in Table S3. Neighboring provinces, including Tianjin,
Hebei, Shandong, Shanxi, and Henan, cooperated with Beijing to increase the effectiveness of these measures. Further, the
Beijing Municipal Government promoted a 2017 revision of the Emergency Plan for Heavy Air Pollution in an effort to
confront future heavy pollution periods. These enhanced measures had demonstrable effects on air quality and $PM_{2.5}$, and
Beijing has since met the targets laid out in the Action Plan. In 2017, the mean concentration of $PM_{2.5}$ was 58 μg m$^{-3}$, with a
year-on-year reduction of 20.5%. This effort was a huge success, and a series of researches associated with the impact of these
clean air actions have been launched (Zheng et al., 2018a;Geng et al., 2019;Li et al., 2019a;Xue et al., 2019;Zhang et al., 2019).
Most of them paid attention to the improvement of air quality and health benefits, the transition of $PM_{2.5}$ chemical composition
and contributors, and the trend of anthropogenic emissions. But nonetheless sufficiently detailed information on ambient VOC
mixing ratios and chemical compositions, as well as variation in emission sources after controls were established, have not
been reported.

Vehicle exhaust, gasoline evaporation, fuel combustion, solvent utilization, and industrial production are the most prevalent
sources of VOCs in Beijing, particularly during hazy days (Wu et al., 2016b;Guo et al., 2012;Sheng et al., 2018). These five

sources were all indicated as controlled objects under the Ten Measures, and SC burning and high-pollution industries were

the most stringent control objects. Compared to 2016, the reduction of civil SC consumption in Beijing exceeded 2 million

tons in 2017. Rural and urban residential sectors contributed 74% and 15% of reduction, respectively, followed by commercial

and public sector (6%) and rural production sector (5%). SC is consumed more, and has greater VOC emissions per unit

combustion, than other fuel types (Fig. 1). Also, a large proportion of civil SC (> 90%) is used for heating in winter. As for

industrial sector, sustained clampdown of the coal-fired boilers was put into action in Beijing from 2013, 99.8% of the boilers

associated with nearly 9 million tons of SC consumption annually were banned and more than half of that were emerged in

2017. All the industrial scattered coal was eradicated by the end of 2017. Therefore, a lot of VOC emissions from SC burning

would be prohibited during the heating season in Beijing.

From 2013 to 2017, 1,992 high-pollution industries were phased out in Beijing, including chemical engineering, furniture

manufacturing, printing, and non-metal mineral product industries. At the municipal level, 11,000 "small, clustered, and

polluting" factories that did not meet efficiency, environmental, or safety standards were either regulated or closed by the end

of 2017, according to the Beijing Municipal Bureau of Economy and Information Technology. Meanwhile, a large part of

high-pollution enterprises (those heavy polluting industries as stipulated by the state environmental protection department)

were removed out of Beijing year by year. The annual variation of designed size enterprises, high-pollution enterprises, and

the annual benefits from industry in Beijing are summarized in Fig. 2. In addition, efforts to increase the quality of gasoline

and diesel fuels began in January 2017; these efforts could lead to a marked decrease in traffic emissions.

Over the course of the study period, control measures were differentially enacted. December 2017 was the most tightly

controlled period, wherein SC burning and substandard coal-fired boilers were forbidden in an effort to meet the targets

identified in the Action Plan. In January 2018, residents were allowed to burn some SC to ensure their well-being. Therefore,

we divided our study into three time periods: non-control (December 2016 – January 2017), strict-control (December 2017),

and eased-control (January 2018). Cold temperatures and a low mixing layer are related to increased emissions and the

accumulation of gaseous pollutants during winter months (Zhang et al., 2015). Thus, this time period represents an ideal opportunity to assess the contributions and effects of various emission sources in Beijing by analyzing ambient VOCs.

**3.2 Ambient VOC concentrations and source contributions**

The implementation of federal and municipal control policies led to significant changes in emission intensity for several sources, which was reflected in ambient VOC concentration characteristics. Fig. 3 shows the ambient VOC concentrations, reported from multiple studies, across seasons in Beijing, where winter generally has the highest values (Liu et al., 2005;Wang et al., 2014;Wei et al., 2018;Li et al., 2019b). Mixing ratios and the chemical composition of VOC groups, as well as the average volume mixing ratios of 91 measured species at PKU, are summarized in Table S4.

Correlations and characteristic ratios between individual VOC species and environmental levels of VOC tracers have been widely used to identify emission sources (Barletta et al., 2005;Liu et al., 2008b). The ratio of benzene and toluene (B:T) can be used to identify VOC sources (Perry and Gee, 1995). An average value of $0.6 \pm 0.2$ (wt/wt) of B:T is characteristic of vehicular emissions in China; this ratio is estimated to be 0.67 in Beijing (Barletta et al., 2005). A higher B:T indicates a greater influence from biomass and/or fossil fuel combustion (Santos et al., 2004;Andreae, 2019). Lower B:T values are related to

solvent utilization due to the abundant use of toluene for painting and printing (Yuan et al., 2010). We estimated B:T (wt/wt) ratios of 0.88, 0.69, and 0.77 for the non-control, strict-control, and eased-control periods, respectively. We suggest that a B:T > 0.67 indicates a significant role of coal combustion for heating, similar to the results reported previous (Wang et al., 2013a). B:T estimations provided in previous researches or observations, as well as in this study, are shown in Fig. 4 (Li et al., 2019b;Li et al., 2015b;Li et al., 2015a;Li et al., 2016c). The B:T reference values for residential coal burning and traffic exhaust and

evaporation are $1.24 \pm 0.20$ and $0.52 \pm 0.06$, respectively (Liu et al., 2008a). B:T values reported for the summer months are closer to the characteristic values for traffic exhaust and evaporation, and those in winter are closer to the characteristic value for coal burning. Results from the strict-control period may represent illegal SC, coal-to-gas, and coal-to-electricity use, and we observed an increase in B:T during the eased-control period relative to the strict-control period.

Analyses of variation in tracers can reflect changes in emission sources. We observed a significant decline in methyl tertiary

butyl ether (MTBE, a common gasoline additive), 2,2-dimethylbutane, 3-methylpentane, methyl cyclopentane, 2-

methylhexane, and 3-methylhexane (all common components in gasoline evaporation and tailpipe exhaust) during the control

period (Chang et al., 2006). Acetonitrile, an inert tracer, can reflect the intensity of biomass burning (Sinha et al., 2014). We

estimated no significant changes in biomass burning by comparing acetonitrile concentrations between the three periods. Freon

113 is typically used to estimate background levels. We found that the concentration of Freon 113 was constant around 0.09 –

0.11 ppbv, indicating a consistent background concentration.

The top 20 most-decreased VOC species after control measures are listed in Table 1. During the strict-control period, tracers

of incomplete burning (e.g., ethylene, acetylene, benzene, styrene, and 1,2-dichloropropane) decreased by > 60%. Tracers of

industrial and vehicle-related sources decreased by 50%, including some chlorinated hydrocarbons, esters and aromatics (Li

et al., 2016c;Hellen et al., 2006;Barletta et al., 2009). We observed a precipitous decline in methacrolein (MACR) and methyl

vinyl ketone (MVK), which are the major oxidation products of isoprene (Xie et al., 2008). Terrestrial vegetation is typically

the main contributor of isoprene in the environment. However, heavy traffic in megacities contributes to a large proportion of

isoprene emissions, particularly after leaf-drop (Song et al., 2007). Ethyl acetate is a widely used industrial solvent, and propene

is characteristic product of internal combustion engines (Scheff and Wadden, 1993). Some aromatics, such as styrene and

benzene, are found in high concentrations in petrochemical plants (Liu et al., 2008a). Benzene, toluene, ethylbenzene, and

xylenes (BTEX) are also major components of vehicle and solvent utilization (Seila et al., 2001). During the eased-control

period, we observed differences in the top declining species and their respective reductions, particularly for tracers of vehicle

exhaust, which had relatively small reductions. Indicator species of oil-refining and fuel burning emissions became more

prevalent during this period, including styrene, C2-C4 alkenes, C3-C10 alkanes, and acetylene. Fuel evaporation is often

indicated by iso-/n-pentane and cyclopentane(Zheng et al., 2018b), both of which showed an obvious decline during the eased-

control period. In both the strict- and eased-control periods, acetylene, a tracer for vehicular and other combustion processes

(Baker et al., 2008), decreased by > 60%. Secondary products from primary anthropogenic VOCs, including ketones and aldehydes, were also reduced (Yuan et al., 2012).

PMF, a receptor-based source apportionment method, was used to estimate temporal variation in source contributions. Eight appropriate factors were determined. Profiles from the literature were referenced in identifying the factor profiles, which were

recognized as: (1) coal burning, (2) fuel oil and gas usage, (3) traffic exhaust, (4) petroleum-related evaporation, (5) VOC-related industry, (6) VOC-product utilization, (7) biomass burning, and (8) transmitted/long-lived species. Modelled source profiles (ppbv ppbv$^{-1}$), together with the relative contributions of individual sources to each parsed species, are shown in Fig. S2. Diurnal and 24 h variation in mixing ratios of all eight sources during the non-control and control periods are also shown in Figs. S2 and S3. Reconstructed diurnal variation and 24 h mixing ratios of controlled sources were lower during the control

periods.

Source contributions (ppbv) and proportions, determined by PMF analyses, are shown in Table 2. Source reduction contributions during strict- and eased- control periods relative to the non-control period are shown in Fig. 5.

PMF is a widely used method to identify emission sources and their contributions (Yuan et al., 2009;Simayi et al., 2020), but its results have some subjectivity and cannot be determined to be absolutely accurate. For this reason, PMF results are usually

mutually corroborated with the actual situation, which is the implementation of control measures in this study: during the strict-control period, coal burning had the greatest reducing contribution (54.33%) relative to the non-control period, followed by petroleum-related evaporation (31.49%) and VOC-related industry (16.25%). During the eased-control period, coal burning contributed 49.33% of total reduction relative to the non-control period, as did petroleum-related evaporation (24.03%) and VOC-related industry (14.26%). The consistent trend between the intensity of the control measures and the proportion of coal

burning and other sources supports the credibility of the PMF results. Another supporting argument is the comparison of the PMF results in this study and PMF analysis from other studies conducted in Beijing, which is summarized by Li et al. (2019b). Comparison of the relative contributions of VOC emission sources in Beijing calculated by the PMF model of this study and results from the other studies during different seasons is listed in Table S7. Other studies show that the fuel combustion, mainly

composed of coal combustion, was the largest VOC contributor in winter. The contribution proportions of fuel combustion in

winter ranged from 45% - 55% (Li et al., 2015a;Yang et al., 2018;Li et al., 2019b), which are even higher than, but still

comparable with that of non-control period in this study (37%). Other studies in Table S7 show that vehicle-related source is

the largest VOC contributor in Beijing, especially in summer and autumn, with the contribution ranged from 50% - 57%, and

33% - 49%, respectively. And the smaller, and comparable contribution of vehicle-related source in winter is reflected among

other studies and this study. During the non-control period, coal burning contributed 37% (33.5 ppbv) of the total ambient

VOCs, far surpassing the contributions of other emission sources, even the combined influence of traffic exhaust and

petroleum-related evaporation (33%, 30.0 ppbv).

The terminal sectors of the burned coal include centralized coal burning (power generation, heat supply, large-scale industrial

boilers), scattered coal burning (rural and urban residential consumption, rural production, commercial and public consumption,

small-scale industrial boilers). Of the overall coal burning in PMF results, SC burning, whose emission factors are far above

centralized coal burning, could contribute much higher emissions than centralized part. Residential and industrial sectors were

the majority part of scattered coal burning in Beijing, both of them contributed more than 90% of all. Generally, residential

SC burning is mostly concentrated during heating season and most of urban families are centrally supplied heating without SC

consumption. It makes residential sectors, especially rural residential sector, more significant in winter than industrial sector.

Monthly distribution of industrial sector is relatively average throughout the year, therefore, SC used in small-scale industrial

boilers and furnaces is much less than residential sectors in winter, Furthermore, higher combustion efficiency and lower VOC

emission factors of industrial sector than civil utilization may make its emissions contribution lower (Bo et al., 2008;Cheng et

al., 2017). A small contribution of coal burning in summer, which is held up by the PMF results of other studies in Table S7,

corroborates the important effect of residential SC burning. Detailed estimation of emissions from coal burning is given in

section 3.3.

### 3.3 Corrected emission inventory for VOCs


EF-based emission inventory is calculated based on statistical activity data and emission factors, both of which have high

degree of uncertainty (Li et al., 2016a), but still it has been widely used to quantify VOC emissions and sort out major emission

sources (Simayi et al., 2019;Wu et al., 2016a). Li et al. (2019b) verified an EF-based VOC emission inventory through ambient

measurements and satellite retrievals, and found that the vehicle-related VOC emissions are reliable and the emissions of

NMHCs are accurate, but the emissions from fuel combustion sources, especially in winter, are largely underestimated. Based

on the EF-based method and the modified source classification system, we established a monthly emission inventory for the

non-control period (December 2016 – January 2017), strict-control period (December, 2017), and eased-control period

(January, 2018). After adding the undocumented SC consumption, which obtained from the existing researches about SC

consumption estimations and control measures for SC burning, errors in coal burning of emission inventory were eliminated,

and the estimation of coal burning contribution of the emission inventory (EI) were much closer to the PMF results (Fig. 6).

The reduction in SC burning in December 2018 was reflected in both the emission inventory and PMF results. Industrial SC

consumption was estimated in the emission inventory but with high uncertainty (Table S5), however, its relatively small

proportion of the total SC consumption in Beijing during heating season would largely reduce its influence on the estimations

of total emissions from SC burning.

Based on the PMF results and the estimated emissions of other sources, coal burning contributed 36.2 ± 10.4 Gg and 14.7 ±

8.6 Gg of anthropogenic VOC emissions in Beijing during the non-control and control periods, respectively. However, at least

80% of the VOC emissions from coal burning were not considered in most existing emission inventory researches (Li et al.,

2019c;Wu and Xie, 2018;Li et al., 2019b). Calculated emissions of SC burning in the corrected emission inventory were 23.7

Gg and 10.5 Gg during the non-control and control periods, respectively, which are comparable to, but lower than, the

estimations calculated using PMF. We note that uncertainty in heating demand and coal quality is a potential reason for these

differences.

The corrected emission inventory had a more comparable proportional distribution to that of the PMF results during the non-control period. However, during the strict- and eased-control periods, we observed poor consistency between the PMF and emission inventory results, particularly for VOC-related industry and petroleum-related evaporation. This could be the result

of neglected factors, such as changes in industrial petrochemical and chemical production during air pollution alerts, and the shut-down of many polluting industries in 2017. We therefore suggest that the emission inventory results for the non-control period are reliable, but VOC-related industry and petrochemical evaporation are overestimated for the control periods.

Total emissions during the non-control period amounted to 82 Gg, and decreased to 59 Gg during the control period. The largest sublevel source, coal burning, had the largest reduction (19 Gg), and the reduction was mostly contributed by rural

residential SC burning. Other sublevel sources also contributed to the overall decline in emissions, such as fuel oil and gas usage (1 Gg), traffic exhaust (4 Gg), petroleum-related evaporation (3 Gg), and VOC-related industry (2 Gg). Emissions from VOC-product utilization and biomass burning did not change significantly between the non-control and control periods. Of the total coal burning during non-control and control period, rural residential SC burning contributed 60% and 68%, respectively; urban residential SC burning contributed 17% and 25%, respectively; industrial SC burning contributed 16% and

0%, respectively; centralized coal burning only contributed 1% and 2%, respectively. Emissions of seven sub-level anthropogenic sources of level 1, major refined sub-contributors of each anthropogenic source, and reduction contribution of each refined sub-contributor from non-control to control period are shown in Fig. 7.

**3.4 SOAP calculations based on PMF results and the corrected emission inventory**

In urban areas, anthropogenic-produced VOCs are major precursors to SOAs (Lin et al., 2009). The formation of SOAs from

VOCs occurs through varied, complex, physical and chemical processes. These are broadly categorized under three main theories within the literature: mechanisms related to photooxidation, nucleation processes, and condensation, gas/particle partition, and heterogeneous reactions (Hallquist et al., 2009;Kroll and Seinfeld, 2008).

Organic gaseous compounds can condense on primary particles, of which the greatest number are within 0.1 – 1 μm. Primary particles rarely coagulate, but do undergo species (including VOC species) exchange in the gas phase. Transformation of



organic vapors to a liquid or solid phase is promoted when the equilibrium vapor pressure is above that of the aerosol surface

(Raes et al., 2000). Generally, molecular clusters tend to evaporate owing to the stronger Kelvin effect, but fulminic nucleation

will occur under suitable conditions. Zhang et al. (2004) suggested that nucleation (new particle formation, NPF) is greatly

enhanced by an interaction between organic and sulfuric acids, particularly in an atmosphere polluted by heavy coal burning.

Two types of NPF events, sulfates-dominated and organics-dominated, have been identified on the North China Plain (Ma et

al., 2016). Condensation and self-coagulation begin, and thus promote growth, around 0.1 μm. Continuous growth of

nucleation-mode particles over several days would lead to haze in Beijing, which has more abundant precursors in the

atmosphere (Guo et al., 2014;Wang et al., 2013b). NPF has been recognized as an important process contributing to the

formation of cloud condensation nuclei (CCN), concentrations of which have increased by 0.4 – 6 times in and around Beijing

(Yue et al., 2011). Typically, organic matter contributes significantly to the mass growth that is characteristic of newly formed

SOAs (Pennington et al., 2013). In Beijing, organic matter is likely the dominant chemical contributor facilitating the

conversion of newly formed particles to CCN.

We used the SOAP approach to determine the effectiveness of the air-quality control period as it pertained to a reduction in

VOC emissions. We note that because SOA formation processes are poorly understood, SOAP was computed to understand

the potential for SOA formation from VOC species, but we could not estimate the actual formation under specific atmospheric

conditions. The concentration of $PM_{2.5}$ was reportedly reduced in 2017 (Beijing Municipal Environmental Protection Bureau,

http://www.bjepb.gov.cn/). The observed large reduction in VOCs may have resulted in the reduction of SOAs and thereby

contributed to the reduction in $PM_{2.5}$. SOAP-weighted mass contributions of each VOC source were used to estimate the

influence of precursor emissions on SOAs. SOAP-weighted mass contributions based on PMF results and corrected emission

inventory are shown in Figs. 8 and 9. According to the factor profiles of PMF results, styrene, toluene, benzene and xylene, as

the major contributors of SOAP, were largely attributed by coal burning (about 40% on average), which accounted for 27% of

ambient VOCs but contributed 40% of total SOAP during the whole study period. Besides, according to the corrected emission

inventory, 47% of benzene, 27% of toluene and 10% of xylene were contributed by coal burning as well.

The greatest contributor to SOAP reduction was coal burning, according to the results of both the PMF and the emission inventory, despite differences in their respective raw values. PMF results indicated that coal burning was the greatest

contributor to SOAP at 18.81 μg m$^{-3}$, accounting for 47% during the non-control period. After control measures were enacted, the contribution of coal burning decreased to 4.56 μg m$^{-3}$, and accounted for 30% of the total. The reduction of coal burning contributed approximately 55% of the total reduction. Petroleum-related evaporation and VOC-related industry contributed 25% and 10% of the total SOAP reduction prior to the establishment of control measures, respectively, and emissions from both sources were reduced over the control period.

Emission inventory results indicated that VOC-related industry was the largest contributor to the total SOAP-weighted mass during both the non-control and control periods. SC burning was the next largest contributor during the non-control period, contributing 64% of the total.

**3.5 Evaluation of control policies**

The idea that reducing residential coal burning will improve air quality in Beijing has been well accepted within the scientific

community (Cheng et al., 2016). Liu et al. (2016) proposed that reductions in residential emissions may have greater benefit to air quality in Beijing than reductions from other emitters during the heating season, and that promoting alternative fuels may be an effective solution. Our study confirmed that coal burning is the greatest contributor to VOC emissions (see PMF results), and we inferred that a large proportion of emissions were the result of SC burning during the heating season in non-control periods (see corrected emission inventory). The contributions of identified VOC sources decreased significantly after

the control period, which means that VOC-related control measures were highly effective. The sharp decrease in SC burning between the non-control and control period related to a reduction in the contribution of coal burning to ambient VOC concentrations. Multiple related measures were enacted during the control period, including the prohibition of SC burning in the countryside, deactivation of coal-fired units in thermal power plants, and conversion of decentralized coal-fired boilers to gas-fired boilers. Coal burning increased slightly between the strict- and eased-control periods, reflecting the allowance of

residential SC burning in January 2018. Petroleum-related evaporation in PMF results was sharply reduced by controlling

high-emission vehicles and reducing leakage from petrochemical industries. We accordingly confirmed that the contribution of coal burning to ambient VOC concentrations exceeded that of traffic-related sources prior to the strict-control period, and that coal burning was the greatest contributor to higher VOC concentrations observed in winter. During and after the strict-control period, VOC concentrations decreased and vehicle exhaust became the main contributor again.

The limitations set by federal and municipal governments on coal burning played a significant role in improving air quality during Beijing's winters. According to the statistics of CESY, the total coal consumption of Beijing reached its highest point of over 30 million tons in 2005, then it kept going down to 4.9 million tons in 2017. And the percentage of coal occupied in primary energy consumption dropped sharply from at least 30% to less than 6% in the past decade. It is reported by State Grid Beijing that Beijing had implemented "switching from coal to electricity" project for residential heating in winter since 2003,

and carried out electric heating exceeding 1.2 million families by 2018. Although significant effects have been achieved in Beijing, civil SC consumption is still widely used nationwide.

For Beijing city, we corrected the SC consumption of both civil and industrial sectors, of which emission factors and monthly activity data are presented in Tables S1 and S5, respectively. But for the nationwide SC estimation, the consumption from industrial sector was difficult to count in most areas, and only data of civil sectors were available. Hence, only civil SC (rural

residential, urban residential, rural production, commercial and public sectors) was discussed (Huo et al., 2017). In 2016, total civil SC consumption was 311.4 million tons in China, and near 65% of them came from rural residential sector. Due to the unprecedented air pollution control measures, civil SC consumption in 2017 achieved a reduction of 18.7 million tons, and rural residential sector took up over 74% of all (Tables S8 and S9). BTH and surrounding provinces, including Beijing, Tianjin, Hebei, Shanxi, Shandong, Henan and Neimenggu, contributed 95% of total reduction (17.8 million tons). However, after the

centralized limitation, BTH and surrounding provinces is still areas with the largest civil SC consumption; Heilongjiang, Guizhou, Hunan and Xinjiang are also provinces with large numbers of civil SC consumption (Fig. 10). To achieve global Sustainable Development Goals (SDGs) (Carter et al., 2019), the control of civil SC consumption is a significant topic to improve air quality, and emission reduction of industrial SC combustion is also worth attention.

Petroleum-related evaporation and VOC-related industry were also significant contributors to VOC and SOAP reductions.

Reductions in these sources were mainly driven by restrictions on petrochemical production and shutting down or regulating

high-pollution industries and small factories. We note that vehicle tailpipe exhaust had a relatively small contribution to the

overall reduction in VOCs, indicating that traffic controls for on-road vehicles had a limited impact. The promotion of higher-

quality gasoline and diesel fuels may have contributed to the observed reduction in contribution from petroleum-related

evaporation.

Several initiatives had been proposed to reduce direct coal burning, particularly uncontrolled and inefficient household SC

burning during the heating season. For example, a "switching from coal to natural gas" project was proposed by the Chinese

government to reduce air pollution from coal-fired boilers and associated premature mortality, and its air quality, health, and

climate implications were also assessed in detail (Qin et al., 2017;Qin et al., 2018). Lu et al. (2019) suggested that deploying

coal-bioenergy gasification systems with carbon capture and storage may provide a promising opportunity for China to realize

its carbon mitigation and air-pollution abatement goals simultaneously. These initiatives could contribute to the development

of future policies aimed at supporting sustainable energy transitions, improving urban air quality and protecting public health.

## 4 Conclusion

The mixing ratios and chemical compositions of 91 VOC species were measured during two sequential winters at PKU in

Beijing, China. Three control periods were defined based on the enforcement of air-quality-control measures. The results of

PMF analyses over these three periods are discussed based on the intensities of source emissions and the strength of control

measures during the heating period. A corrected monthly EF-based inventory was compared to the PMF results, and this

comparison suggested that coal burning, which had been identified as a large contributor of primary particles, contributed a

large proportion of the total VOC emissions and total SOA potential during the winter months in Beijing. Coal burning was

the largest contributor during the non-control period, exceeding vehicle-related sources, and SC burning accounted for > 98%

of all coal burning emissions. Vehicle exhaust was the greatest contributor during the strict-control period, when regulation on

SC burning were enforced. The contribution from coal burning increased again during the eased-control period as SC was allowed as a residential heating source. On all accounts, differences in the results of PMF analyses and the emission inventory in preceding studies were mainly due to a gap in the estimation of SC burning, which had not previously been quantified. Mainly affected by the enhanced limitation on SC consumption, the VOC emissions in Beijing has been significantly cut down

during heating seasons, and sustained shifting of cleaner energy use-patterns will help to further improve air quality. Moreover, passenger car exhaust, petrochemical manufacturing, gas stations, traffic evaporation, traffic equipment manufacturing, painting, and electronics manufacturing are also contributors to ambient VOCs, which need to be focused on future emission reduction strategies and targets in Beijing.

Although the SC burning has been cut down in Beijing, our detailed discussion of the effect of control measures in this study

can provide valuable reference for the haze control in other regions where the civil SC consumption prevail, especially in winter. Those regions which rely on SC and biomass as primary civil heating fuels nowadays, should gradually promote clean energy transformation in accordance with the level of local economic development and the living standard of local residents (Carter et al., 2019). The improvement of fossil fuel combustion efficiency and popularization of clean energy use, will be propitious to air quality and people's health and help to achieve sustainable development in China.


*Data availability.* The datasets that include the measurements and emissions can be accessed by contacting the corresponding author (Shaodong Xie; sdxie@pku.edu.cn).

*Author contributions.* SDX designed the study, YQS performed the data analysis and wrote the paper. ZYX contributed to the

development of the emission inventory. MS participated in data collection. JL assisted with the online measurements. All authors assisted with interpretation of the results and the writing of the paper.



*Competing interests.* The authors declare that they have no conflict of interest.

*Acknowledgements.* This work was supported by the National Air Pollution Prevention Joint Research Center of China for "The research of characteristics, emission reduction and regulatory system of volatile organic compounds in key sectors" (grant numbers DQGG0204)

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





**Figure and Table captions:**

**Table 1. The top 20 major VOC species with the highest decreasing ratios compared with non-control period.**

**Table 2. Source contributions to ambient VOCs concentration (ppbv) and their contributions to total reduction compared with non-control period (ratio) derived by PMF analysis.**

**Figure 1. VOC emission factor (g kg$^{-1}$ fuel) and consumption (million tons) of different fuel types, (a) for consumption of non-control period (Dec.2016-Jan.2017) and (b) for consumption of control period (Dec.2017-Jan.2018).**

**Figure 2. Quantities and variations of designed size enterprises and high-pollution enterprises from late 2012 to early 2018 (bridge figure), and industrial added value from 2012 to 2018 (line).**

**Figure 3. Ambient VOCs concentration (ppbv) in different seasons of Beijing.**

**Figure 4. The ratio of benzene and toluene in different seasons (previous studies) and different control periods (this study).**

**Figure 5. Different sources contribution of strict- and eased- control period to total reduction compared with non-control period (ratio) derived by PMF analysis.**

**Figure 6. The comparison of PMF results and corrected emission inventory of different control periods.**

**Figure 7. Emissions (Gg) of seven sub-level anthropogenic sources of level 1; major refined sub-contributors of each anthropogenic source; and reduction contribution of each refined sub-contributor from non-control to control period.**

**Figure 8. SOAP-weighted mass contributions of different sources and their contribution to total reduction (%) based on PMF results.**

**Figure 9. SOAP-weighted mass contributions of different sources and their contribution to total reduction (%) based on emission inventory.**

**Figure 10. Map of provinces civil SC consumption in mainland China in 2017 and proportion of different terminal sectors. (Lack of data in Tibet)**



**Table 1. The top 20 major VOC species with the highest decreasing ratios compared with non-control period.**

| Species | Decreasing ratio (strict control) | | Decreasing ratio (eased control) |
|---|---|---|---|
| methacrolein | 79.1% | cyclohexane | 77.6% |
| methyl ethyl ketone | 72.2% | 1,2-dichloropropane | 73.2% |
| benzene | 67.5% | styrene | 71.1% |
| styrene | 66.2% | methyl vinyl ketone | 70.5% |
| 1,2-dichloropropane | 64.6% | 1,1-dichloroethane | 70.2% |
| methyl vinyl ketone | 63.8% | acrolein | 69.9% |
| acrolein | 62.5% | benzene | 68.3% |
| acetylene | 62.1% | m/p-xylene | 67.5% |
| ethylene | 61.9% | ethylene | 66.6% |
| cis-2-butene | 61.5% | cis-2-butene | 65.6% |
| m/p-xylene | 60.8% | toluene | 65.4% |
| 1,4-dichlorobenzene | 60.8% | isoprene | 64.7% |
| propanal | 60.3% | o-xylene | 64.7% |
| toluene | 59.5% | propylene | 64.3% |
| 1,1-dichloroethane | 59.3% | acetylene | 64.2% |
| o-xylene | 57.6% | ethylbenzene | 63.8% |
| isoprene | 57.0% | propanal | 63.6% |
| acetone | 56.6% | methyl ethyl ketone | 63.4% |
| propylene | 56.3% | 3-methyl pentane | 62.1% |
| ethylbenzene | 56.2% | acetone | 61.3% |



**Table 2. Source contributions to ambient VOCs concentration (ppbv) and their contributions to total reduction compared with non-control period (ratio) derived by PMF analysis.**

| Source | non-control period | | strict-control period | | eased-control period | |
|---|---|---|---|---|---|---|
| | Source contribution | Percentage | Source contribution | Percentage | Source contribution | Percentage |
| Coal burning | 33.51 | 37.33% | 8.15 | 18.91% | 8.09 | 21.17% |
| Fuel oil and gas usage | 3.28 | 3.66% | 3.06 | 7.10% | 4.98 | 13.03% |
| Traffic exhaust | 11.86 | 13.22% | 8.68 | 20.16% | 6.62 | 17.31% |
| Petroleum-related evaporation | 18.15 | 20.22% | 3.45 | 8.02% | 5.77 | 15.09% |
| VOC-related industry | 12.32 | 13.73% | 4.74 | 11.00% | 4.98 | 13.02% |
| VOC-product utilization | 3.99 | 4.44% | 6.58 | 15.29% | 1.77 | 4.62% |
| Biomass burning | 1.43 | 1.60% | 1.28 | 2.98% | 1.78 | 4.66% |
| Transmitted/long-lived species | 5.2 | 5.80% | 7.13 | 16.54% | 4.24 | 11.09% |
| total | 89.75 | 100.00% | 43.07 | 100.00% | 38.23 | 100.00% |

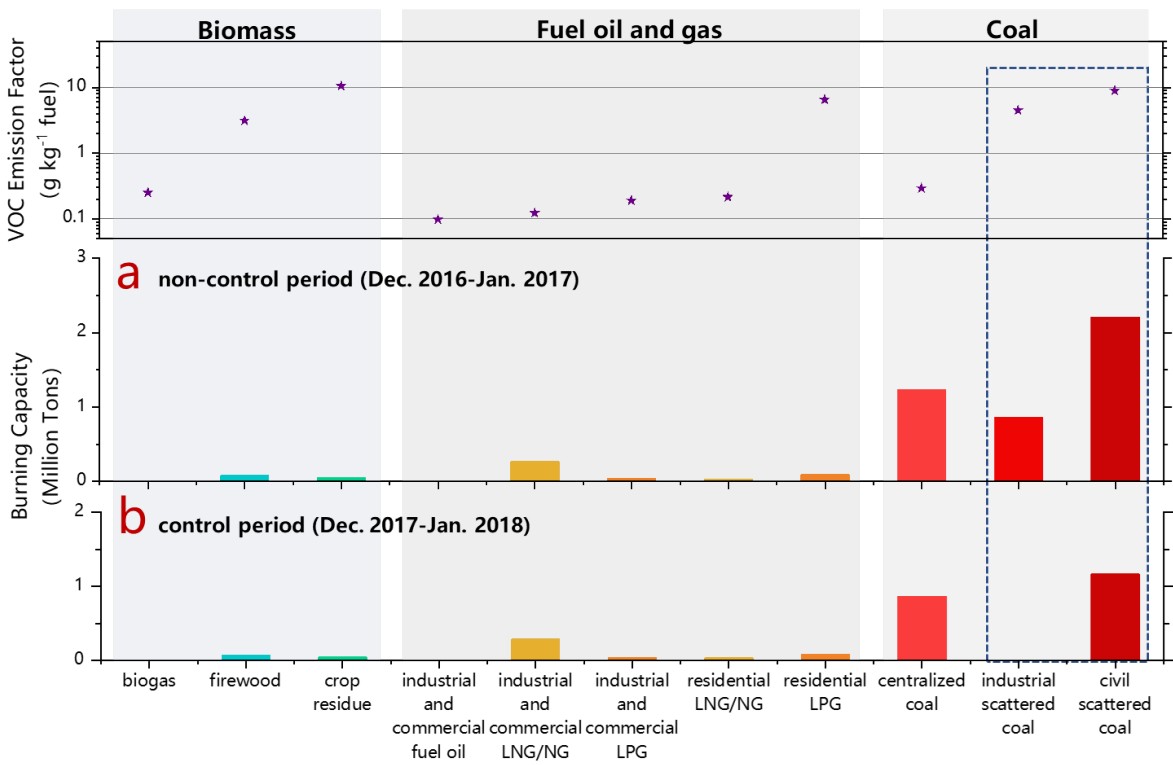

**Figure 1. VOC emission factor (g kg$^{-1}$ fuel) and consumption (million tons) of different fuel types, (a) for consumption of non-control period (Dec.2016-Jan.2017) and (b) for consumption of control period (Dec.2017-Jan.2018).**

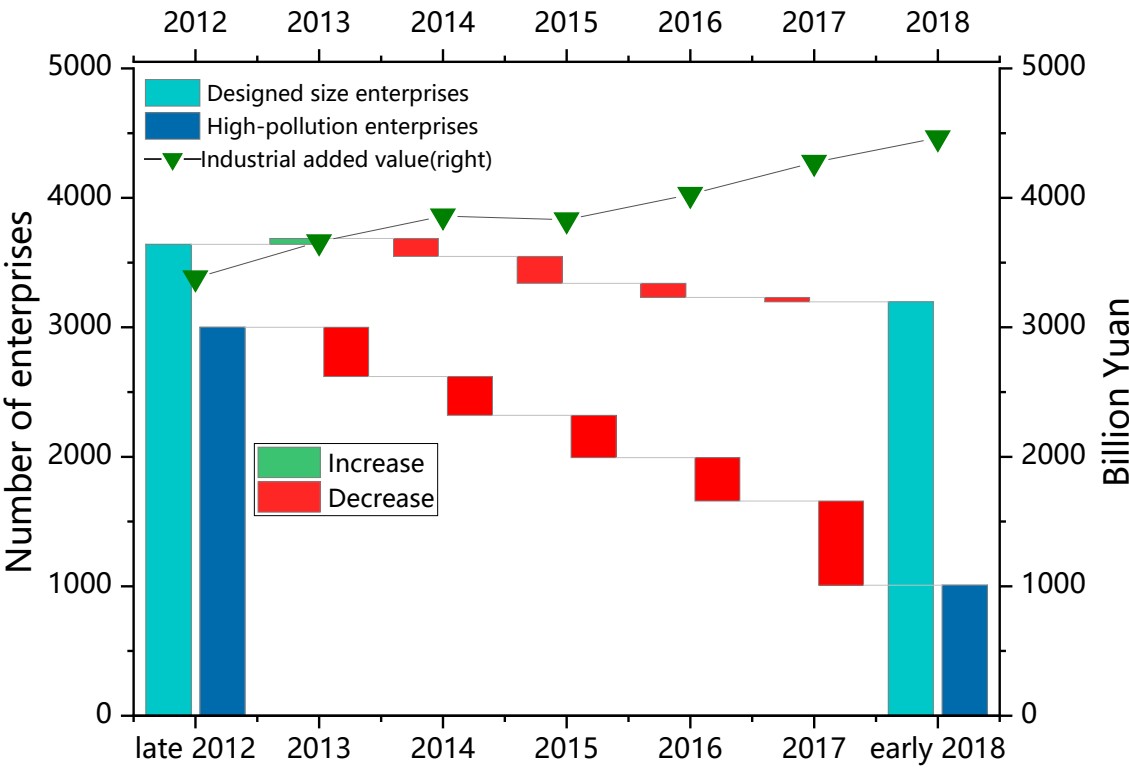

750

**Figure 2. Quantities and variations of designed size enterprises and high-pollution enterprises from late 2012 to early 2018 (bridge figure), and industrial added value from 2012 to 2018 (line).**



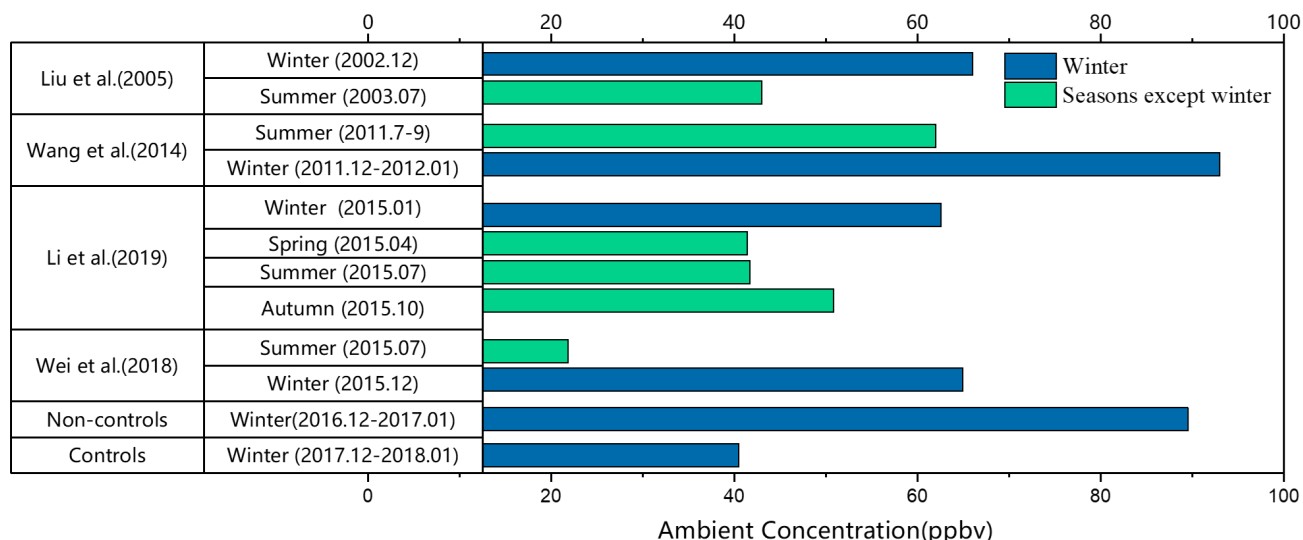

**Figure 3. Ambient VOCs concentration (ppbv) in different seasons of Beijing.**





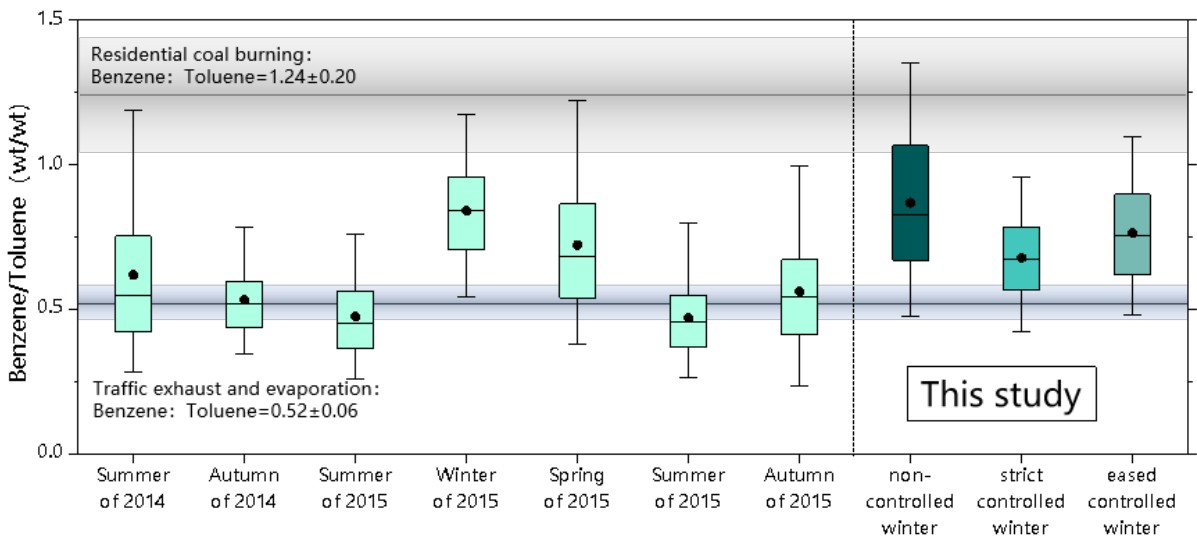

755

**Figure 4. The ratio of benzene and toluene in different seasons (previous studies) and different control periods (this study).**





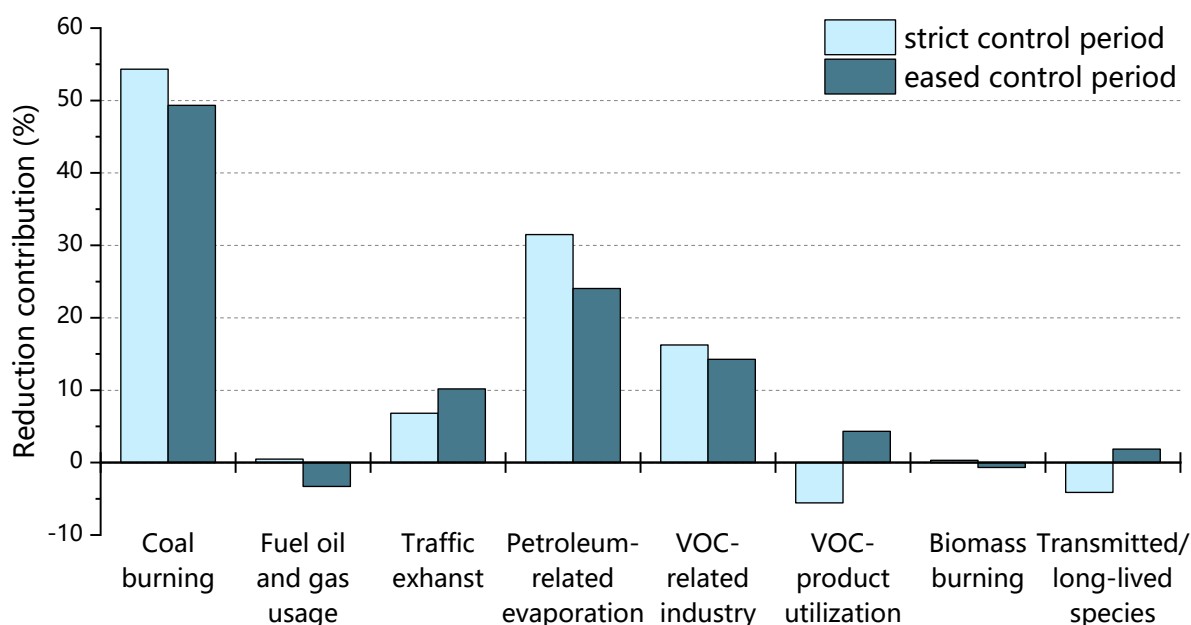

**Figure 5. Different sources contribution of strict- and eased- control period to total reduction compared with non-control period (ratio) derived by PMF analysis.**





**Figure 6. The comparison of PMF result and corrected emission inventory of different control periods.**

760





Figure 7. Emissions (Gg) of seven sub-level anthropogenic sources of level 1; major refined sub-contributors of each anthropogenic source; and reduction contribution of each refined sub-contributor from non-control to control period.





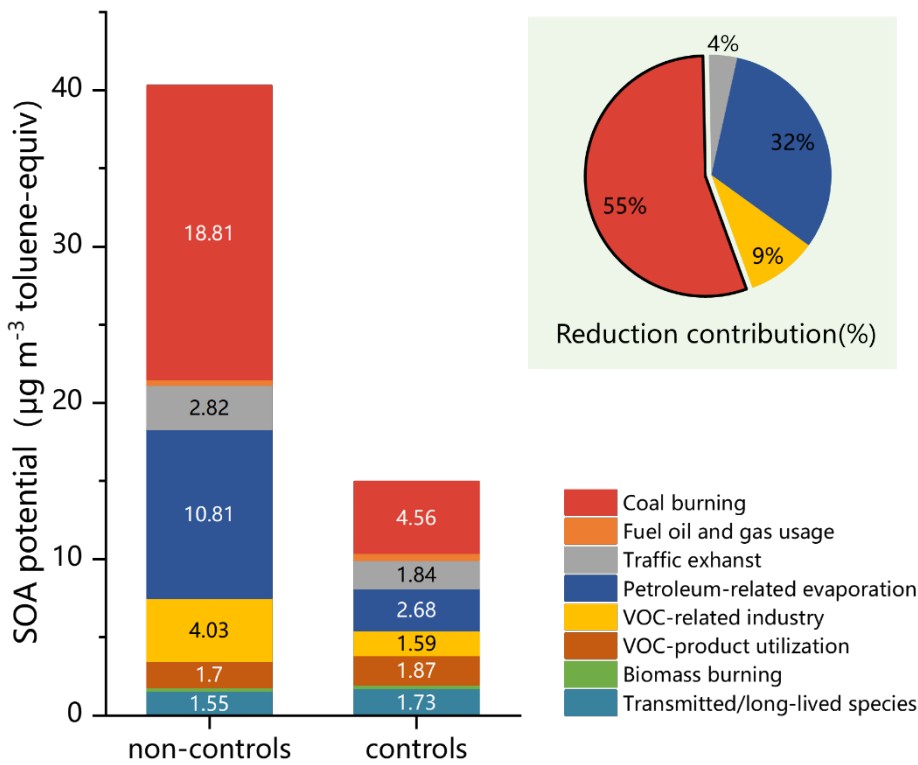

765

**Figure 8. SOAP-weighted mass contributions of different sources and their contribution to total reduction (%) based on PMF results.**



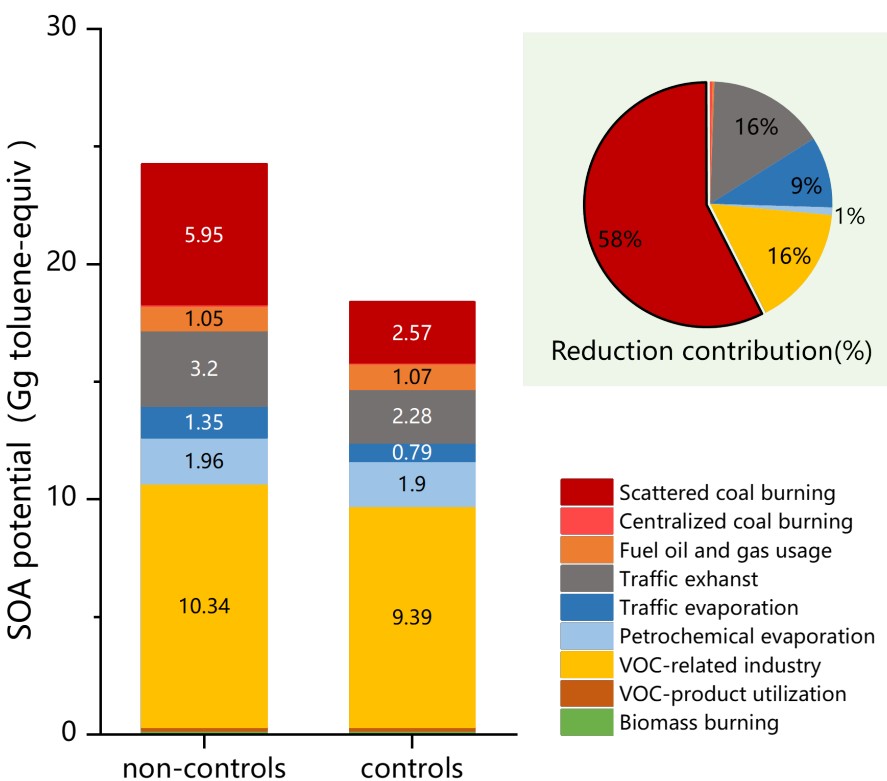

**Figure 9. SOAP-weighted mass contributions of different sources and their contribution to total reduction (%) based on emission inventory.**





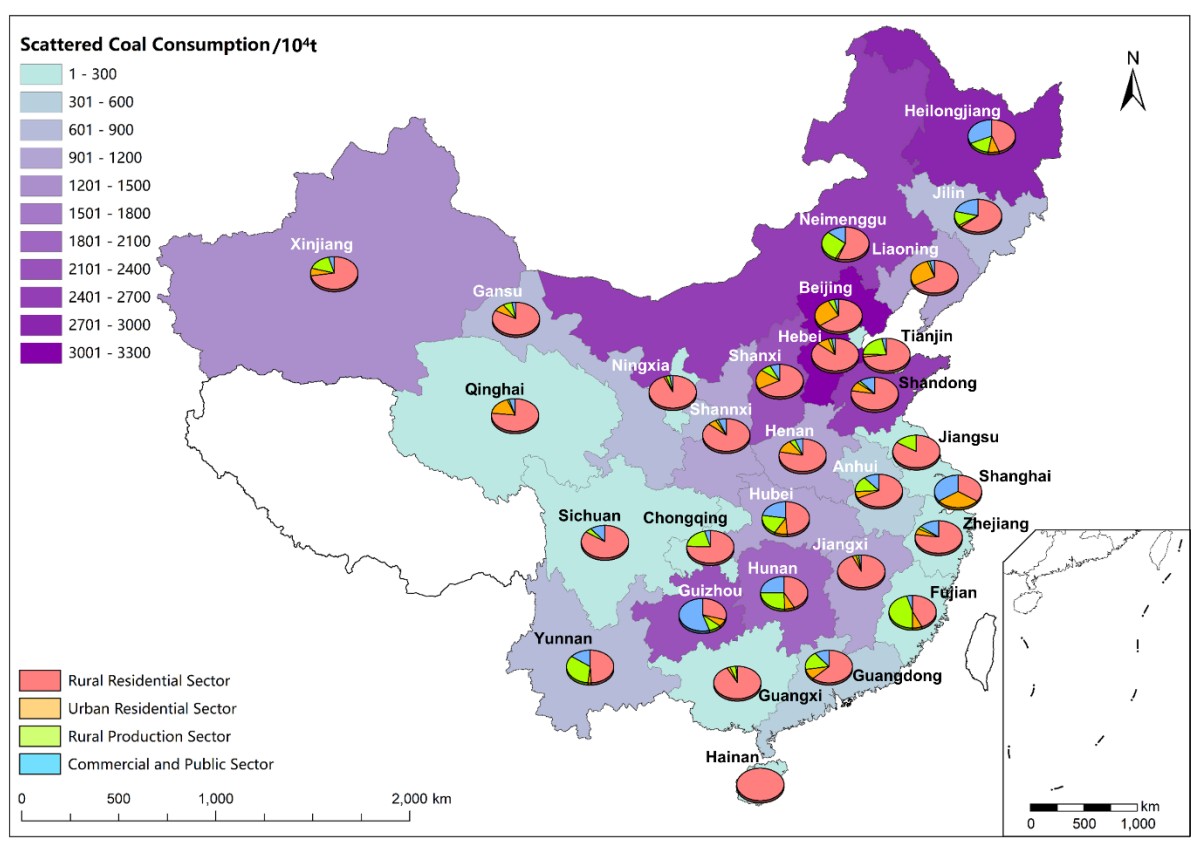

**Figure 10. Map of provinces civil SC consumption in mainland China in 2017 and proportion of different terminal sectors. (Lack of data in Tibet)**