# Peer review of "Scattered coal is the largest source of ambient volatile organic compounds during the heating season in Beijing"

_Atmospheric Chemistry and Physics, 2020_

## Referee Comment (RC1) · Anonymous Referee #2 · 24 May 2020

The authors have addressed all my comments; I suggest accepting it.

---

## Referee Comment (RC2) · Anonymous Referee #1 · 27 May 2020

General comments

Shi et al. present VOC mixing ratios recorded in Beijing. Using positive matrix factorisation they identify coal burning as the largest contributor to VOC emissions prior to emissions controls being imposed. A decrease in scattered coal burning was a major factor in the observed decrease in VOC emissions and therefore secondary organic aerosol formation potential after controls were imposed. The impact of emission controls on air quality are currently of great interest. I recommend publication once the specific comments outlined below have been addressed.

Specific comments

[Figure]

The winter haze in Beijing is often driven by meteorology (e.g. Jia et al., 2008). Given the two month measurement period I would not expect this to have a great effect but a short discussion of the meteorology in each measurement period would demonstrate that the drop in mixing ratio observed in the control period was not caused by different meteorological conditions.

The authors report "total VOC", when referring to sum measured VOC. When this term is first used some discussion of the limitations of the measurement technique and which VOC species may not be included here is required (e.g. methanol and ethanol).

"Mixing ratio" and "concentration" are used interchangeably throughout (e.g. lines 226 and 229). Where referring to values in ppb "mixing ratio" should be used.

Line 2: Define "heating season" in the abstract

Line 85: Figure S1 does not do much to allow the reader to learn more about the site location. I suggest zooming in further on the right hand image and adding details to the map.

Line 91: Describe the "rigorous" QA and QC procedures applied

Line 102: What was the signal to noise threshold for the compounds not included and why was this value chosen?

Line 106: By what criteria were VOCs categorised as "bad"

Line 113: Systematic literature reviews generally follow a clearly defined protocol. I suggest defining the criteria used in the review or deleting "systematic"

Line 129: When stating "Many studies" please provide citations

Line 202: Add citation for the reduction of civil SC in Beijing exceeding 2 million tons or describe where this figure is from.

Figs 1 and 2: Where are these data from?

Line 214: Define "designed size enterprise"

Fig 2: Does the industrial added value refer to just the designed size enterprises and high pollution enterprises or is this a Beijing total?

Section 3.2. A discussion of the VOC mixing ratios recorded in this study at the start of this section would provide context for the % reductions reported. Without knowing how high mixing ratios were it's hard for the reader to interpret the % reductions. This could be achieved by adding mixing ratios to table 1.

Fig 3. See earlier comment, which VOCs are included in this figure? If reporting pbb suggest changing the "concentration" to "mixing ratio" in the caption

Fig 4: Add names and citations for previous studies shown.

Minor comments:

Line 62: Suggesting changing "400×104" to "4×106"

Figure S4: Label figure to show control and non-control periods (i.e. which is top and which is bottom).

Lines 204 – 207: This sentence isn't very clear, suggest rewording

Line 212: Suggest changing "part" to "number"

Line 251: "top 20 most decreased VOC species" is a slightly confusing term. This term is used throughout this section. I suggest changing to "the 20 VOC species which declined the most following emissions controls".

Line 252: Suggest adding "mixing ratios of the" after "period,"

References

Jia Y.T., Rahn K.A., He K.B., Wen T.X., and Wang Y.S.: A novel technique for quantifying the regional component of urban aerosol solely from its sawtooth cycles, Journal of Geophysical Research-Atmospheres, 113, D21309, 2008.

---

## Author Comment (AC1) · 8 Jun 2020

Thank you for your letter dated 24 May, 2020. We were pleased to know that our work was suggested as acceptable for publication in Atmospheric Chemistry and Physics after addressing your comments. We thank you again for your constructive criticisms that have helped us to improve our manuscript.
* * *

---

## Author Response (AR2)

**List of Responses**

Dear Editor and Reviewers,

I really appreciate for the time and effort that you have put into reviewing this manuscript. All your comments and suggestions have enabled us to greatly improve our work. We have studied comments carefully and have made itemized responses in below. Attached please find the revised version with all the changes highlighted by using red font.

Responds to the reviewer's comments:

**Anonymous Referee #1:**

**General Comments:**

Shi et al. present VOC mixing ratios recorded in Beijing. Using positive matrix factorization, they identify coal burning as the largest contributor to VOC emissions prior to emissions controls being imposed. A decrease in scattered coal burning was a major factor in the observed decrease in VOC emissions and therefore secondary organic aerosol formation potential after controls were imposed. The impact of emission controls on air quality are currently of great interest. I recommend publication once the specific comments outlined below have been addressed.

**Specific Comments:**

1. The winter haze in Beijing is often driven by meteorology (e.g. Jia et al., 2008). Given the two month measurement period I would not expect this to have a great effect but a short discussion of the meteorology in each measurement period would demonstrate that the drop in mixing ratio observed in the control period was not caused by different meteorological conditions.

**Response: Accepted.** Thanks for point out the defect in our manuscript. We have added a brief description about the meteorology in section 3.1. The details are given as follows:

Line 258-264: In addition, we discussed several meteorological parameters in Beijing during the study period (Table S4). Temperature, wind speed and wind direction data were acquired from National Oceanic and Atmospheric Administration (https://www.noaa.gov/), snowfall and relative humidity data were from China Meteorological Administration (http://www.cma.gov.cn/). Little snowfall, low speed ($\leq$ 3 m/s) winds and northerly winds were dominant during both the non-control and control periods, and the differences of average temperature and average wind speed between the two periods were 1.2 °C and 0.7 m/s, respectively, indicating the minor influence from meteorological variability on the change of VOC mixing ratios.

**Table S4. Summary of average meteorological parameters during the non-control and control periods.**

| Meteorological Parameters | Non-control Dec.2016 - Jan.2017 | Control Dec.2017 – Jan.2018 |
|---|---|---|
| Temperature (°C) | -1.4 | -2.6 |
| Wind Speed (m/s) | 2.8 | 3.5 |
| Wind Speed ≤ 3 (proportion, %) | 74 | 62 |
| Northerly Winds (proportion, %) | 67 | 63 |
| Snowfall (days) | 4 | 2 |
| Relative Humidity (%) | 63 | 41 |

[a] Meteorological data were all measured at a time interval of 0.5 h.

[b] Northerly wind includes NW, NNW, N, NNE, NE.

2. The authors report "total VOC", when referring to sum measured VOC. When this term is first used some discussion of the limitations of the measurement technique and which VOC species may not be included here is required (e.g. methanol and ethanol).

**Response: Accepted.** We apologize for the unclarity in the manuscript. We have added some discussions about the limitations of the measurement technique and pointed VOC species not included in the study, see section 2.2. The details are given as follows:

Line 93-102: Calibration curves were performed at six mixing ratios from 0.2 to 8 ppbv for each compound before and after sample analyses by bubbling a series of external calibrating gases. Two types of gases were used: a Photochemical Assessment Monitoring Stations (PAMS) ozone precursor series (mixture of 57 NMHCs), and a gas series customized by the PKU National Key Laboratory (a mixture of 55 oxygenated VOCs and halocarbons). In addition, internal calibrating gases were pumped into the GC-MS system once sampling or calibrating to reduce instrumental error. All four calibrating gases were obtained from Linde Electronics and Specialty Gases, USA. $R^2$ (coefficient of determination) values of eligible calibration curves are > 0.99. VOC species can be quantified only if they have eligible calibration curves. Several VOC species also cannot be quantified because their mixing ratios were below method detection limit (MDL). Finally, a total of 91 VOC species were quantified (Table S5), not including formaldehyde, acetaldehyde, and alcohols.

3. "Mixing ratio" and "concentration" are used interchangeably throughout (e.g. lines 226 and 229). Where referring to values in ppb "mixing ratio" should be used.

**Response: Accepted.** Thanks for pointing out the error and we have checked through the manuscript carefully to ensure the reasonable using of "mixing ratio" and "concentration".

**4. Line 2: Define "heating season" in the abstract.**

**Response: Accepted.** The definition of heating season was adjusted in the introduction, and added in the abstract. The definition is "the cold season when fossil fuel is burned for residential heating".

**5. Line 85: Figure S1 does not do much to allow the reader to learn more about the site location. I suggest zooming in further on the right hand image and adding details to the map.**

**Response: Accepted.** According to your justified comment, we have redrawn Figure S1 as below. Sources of base maps were given in the caption.

[Figure]

**Figure S1. The location of (a) Beijing in China (http://bzdt.ch.mnr.gov.cn/) and (b) Peking University (PKU) in Beijing (http://openstreetmap.org/); and (c) the surroundings of the sampling site at PKU (https://www.mapbox.com/).**

6. Line 91: Describe the "rigorous" QA and QC procedures applied.

**Response: Accepted.** Thank you for pointing out our omission. We have explained the "rigorous" QA and QC procedures in section 2.2 as shown below:

Line 103-108: We applied rigorous quality-assurance (QA) and quality-control (QC) procedures which included three main parts. First, daily maintenance and monitoring of the online GC–MS/FID system were performed to ensure the normal operation of instrument. Second, periodic supplement and replacement of consumable items were performed at least every 10 days to ensure the operation of automatic sampling and measuring. Third, periodic calibrations were performed every 5 days, and the calibration curve results of each target species with < 10% variation were considered acceptable relative to the actual values.

7. Line 102: What was the signal to noise threshold for the compounds not included and why was this value chosen? Line 106: By what criteria were VOCs categorised as "bad"?

**Response: Accepted.** In the revised version of our manuscript, we added the signal to noise threshold for VOC species omitted form PMF analysis and criteria for species defined as "bad", and we gave some explanation about them in section 2.3.

Line 118-120: According to the input files, signal-to-noise ratio (S:N) was calculated for each species, and only mixing ratios that exceed the uncertainty contribute to the signal portion in the PMF version we used. Signal is the difference between mixing ratio and uncertainty and noise is the uncertainty value.

Line 122-127: A species is not appropriate for source apportionment if it is undetectable (< MDL) in most of the samples or its mixing ratio always below the uncertainty (signal = 0). Therefore, VOC species that were below the MDL in > 50% of samples or that showed S:N = 0 were categorized as "bad" directly. Other species were categorized based on detailed knowledge of the sources, sampling, and analytical uncertainties (Reff et al., 2007). For species without detailed information, as mentioned in PMF user guide, we conservatively categorized them as "good" if S:N > 1, "Bad" if S:N < 0.5 and "Weak" if S/N < 1 but > 0.5.

8. Line 113: Systematic literature reviews generally follow a clearly defined protocol. I suggest defining the criteria used in the review or deleting "systematic".

**Response: Accepted.** Thank you for pointing out our inappropriate expression. We browsed most of the literature about anthropogenic VOC emission inventory but the procedure is subjective and cannot be described explicitly, therefore, we deleted "systematic" as you suggested. (Line 134)

9. Line 129: When stating "Many studies" please provide citations.

**Response: Accepted.** We have provided the citations when stating "many studies" as below:

Line 153-154: Many studies have estimated the SC consumption recent years (Liu et al., 2016;Cheng

et al., 2017;Huo et al., 2017;Peng et al., 2019), but few of that have estimated SC reductions in 2017 compared to 2016.

10. Line 202: Add citation for the reduction of civil SC in Beijing exceeding 2 million tons or describe where this figure is from? Figs 1 and 2: Where are these data from?

**Response: Accepted.** Thank you for pointing out our omission. Fig 1 and Line 202: for fuel consumption, all relevant data were directly given in or deduced from China Energy Statistical Yearbook (CESY) and COALCAP report; for emission factors, exact values and their detailed references were provided in Table S1. Fig 2: industrial added value and quantity of industries above designated size were from Beijing Municipal Bureau Statistics (BBS, http://tjj.beijing.gov.cn/); quantity of high-pollution industries was from Beijing Municipal Ecology and Environment Bureau (BMEE, http://sthjj.beijing.gov.cn/). We have added these citations as you suggested as below:

Line 227-228: Compared to 2016, the reduction of civil SC consumption in Beijing exceeded 2 million tons in 2017 (CESY, 2017 – 2018; COALCAP reports, 2017 – 2018).

Line 234-238: As shown in Fig. 1, compared with other fuel types, SC is consumed more in winter (CESY, 2017 – 2018; COALCAP reports, 2017 – 2018), and has greater VOC emissions per unit combustion (see Table S1 for details).

Line 247-249: The annual variations of industries above designated size (BBS), high-pollution industries (BMEE), and the annual benefits from industry (BBS) in Beijing are summarized in Fig. 2.

11. Line 214: Define "designed size enterprise"

**Response: Accepted.** To make it clear, we referred the expression given in the statistical yearbook and fixed "designed size enterprise" as "industry above designated size". Industry above designated size is defined as industry with an annual main business income of more than 20 million yuan. (Line 249-250)

12. Fig 2: Does the industrial added value refer to just the designed size enterprises and high pollution enterprises or is this a Beijing total?

**Response: Accepted.** The industrial added value provided here is a Beijing total, which is the sum of added value of all industrial units in Beijing. (Line 250)

13. Section 3.2. A discussion of the VOC mixing ratios recorded in this study at the start of this section would provide context for the % reductions reported. Without knowing how high mixing ratios were it's hard for the reader to interpret the % reductions. This could be achieved by adding mixing ratios to table 1.

**Response: Accepted.** Thank you for underlining this deficiency. The modified Table 1 has replaced the old version in the manuscript, and we gave it as below:

**Table 1. The 20 VOC species which declined the most following emissions controls during strict-control and eased-control periods.**

| Species | non-control (ppbv) | strict-control (ppbv) | Decreasing ratio (%) | Species | non-control (ppbv) | eased-control (ppbv) | Decreasing ratio (%) |
|---|---|---|---|---|---|---|---|
| methacrolein | 1.18 | 0.25 | 78.8% | cyclohexane | 0.11 | 0.03 | 72.7% |
| methyl ethyl ketone | 0.78 | 0.22 | 71.8% | 1,2-dichloropropane | 0.51 | 0.14 | 72.5% |
| benzene | 3.27 | 1.06 | 67.6% | acrolein | 0.14 | 0.04 | 71.4% |
| styrene | 0.37 | 0.12 | 67.6% | 1,1-dichloroethane | 0.17 | 0.05 | 70.6% |
| 1,2-dichloropropane | 0.51 | 0.18 | 64.7% | styrene | 0.37 | 0.11 | 70.3% |
| acrolein | 0.14 | 0.05 | 64.3% | methyl vinyl ketone | 0.50 | 0.15 | 70.0% |
| methyl vinyl ketone | 0.50 | 0.18 | 64.0% | benzene | 3.27 | 1.03 | 68.5% |
| acetylene | 8.98 | 3.40 | 62.1% | m/p-xylene | 0.85 | 0.28 | 67.1% |
| ethylene | 12.07 | 4.60 | 61.9% | cis-2-butene | 0.09 | 0.03 | 66.7% |
| m/p-xylene | 0.85 | 0.33 | 61.2% | isoprene | 0.12 | 0.04 | 66.7% |
| propanal | 0.53 | 0.21 | 60.4% | ethylene | 12.07 | 4.03 | 66.6% |
| 1,4-dichlorobenzene | 0.20 | 0.08 | 60.0% | toluene | 3.63 | 1.26 | 65.3% |
| toluene | 3.63 | 1.47 | 59.5% | o-xylene | 0.65 | 0.23 | 64.6% |
| 1,1-dichloroethane | 0.17 | 0.07 | 58.8% | propylene | 2.10 | 0.75 | 64.3% |
| isoprene | 0.12 | 0.05 | 58.3% | acetylene | 8.98 | 3.21 | 64.3% |
| o-xylene | 0.65 | 0.28 | 56.9% | propanal | 0.53 | 0.19 | 64.2% |
| acetone | 6.37 | 2.77 | 56.5% | methyl ethyl ketone | 0.78 | 0.28 | 64.1% |
| ethylbenzene | 0.96 | 0.42 | 56.3% | ethylbenzene | 0.96 | 0.35 | 63.5% |
| propylene | 2.10 | 0.92 | 56.2% | 3-methyl pentane | 0.61 | 0.23 | 62.3% |
| cis-2-butene | 0.09 | 0.04 | 55.6% | acetone | 6.37 | 2.46 | 61.4% |

14. Fig 3. See earlier comment, which VOCs are included in this figure? If reporting ppb suggest changing the "concentration" to "mixing ratio" in the caption

**Response: Accepted.** Thank you for your suggestion. We have changed the "concentration" to "mixing ratio" in Fig. 3, as shown below.

[Figure]

**Figure 3. Ambient VOC mixing ratios (ppbv) in different seasons of Beijing.**

15. Fig 4: Add names and citations for previous studies shown.

**Response: Accepted.** Thank you for pointing out our omission. Names and citations have been added in Fig. 4 to clarify their sources as shown below.

[Figure]

**Figure 4. The ratios of benzene and toluene in different seasons (previous studies) and different control periods (this study).**

**Minor comments:**

1. Line 62: Suggesting changing "400×104" to "4×106"

**Response: Accepted.** We have made change as you suggested. (Line 62)

2. Figure S4: Label figure to show control and non-control periods (i.e. which is top and which is bottom).

**Response: Accepted.** We have added labels in Figure S4 to give a clear indication.

[Figure]

**Figure S4: Full-time variations in mixing ratios of eight sources in Beijing during the (a) non-control and (b) control periods.**

3. Lines 204 – 207: This sentence isn't very clear, suggest rewording

**Response: Accepted.** This sentence has been reworded. The details are given as follows:

Line 234-238: A large proportion of civil SC (> 90%) is used for heating in winter. As shown in Fig. 1, compared with other fuel types, SC is consumed more in winter (CESY, 2017 – 2018; COALCAP Reports, 2017 – 2018), and has greater VOC emissions per unit combustion (see Table S1 for details). As for industrial SC burning, sustained clampdown of the coal-fired boilers was put into action in Beijing from 2013, and 99.8% of them had been banned before late 2017. These banned boilers contributed nearly 9 million tons of SC reductions and more than half of them were eradicated in 2017.

4. Line 212: Suggest changing "part" to "number"
Response: **Accepted.** We have made change as you suggested. (Line 245)

5. Line 251: "top 20 most decreased VOC species" is a slightly confusing term. This term is used throughout this section. I suggest changing to "the 20 VOC species which declined the most following emissions controls".
Response: **Accepted.** We have modified this expression throughout the section according to the comment.

6. Line 252: Suggest adding "mixing ratios of the" after "period,"
Response: **Accepted.** We have added the content as you suggested. (Line 294)

**Response: Accepted.**

Revised version: Industry above designated size is defined as industry with an annual main business income of more than 20 million yuan.

We have studied your comments carefully and have made technical corrections as you suggested.

**List of all relevant changes made in the manuscript**

[revised manuscript text omitted]